# The Use of Hydrogel Dressings in Sulfur Mustard-Induced Skin and Ocular Wound Management

**DOI:** 10.3390/biomedicines11061626

**Published:** 2023-06-02

**Authors:** Fanny Caffin, David Boccara, Christophe Piérard

**Affiliations:** 1Institut de Recherche Biomédicale des Armées, 1 Place du Général Valérie André, 91220 Brétigny-sur-Orge, France; christophe.pierard@orange.fr; 2Hôpital Saint-Louis, 1 Avenue Claude Vellefaux, 75010 Paris, France; david.boccara@aphp.fr

**Keywords:** sulfur mustard, hydrogels, wound dressings, burn, skin, eye

## Abstract

Over one century after its first military use on the battlefield, sulfur mustard (SM) remains a threatening agent. Due to the absence of an antidote and specific treatment, the management of SM-induced lesions, particularly on the skin and eyes, still represents a challenge. Current therapeutic management is mainly limited to symptomatic and supportive care, pain relief, and prevention of infectious complications. New strategies are needed to accelerate healing and optimize the repair of the function and appearance of damaged tissues. Hydrogels have been shown to be suitable for healing severe burn wounds. Because the same gravity of lesions is observed in SM victims, hydrogels could be relevant dressings to improve wound healing of SM-induced skin and ocular injuries. In this article, we review how hydrogel dressings may be beneficial for improving the wound healing of SM-induced injuries, with special emphasis placed on their suitability as drug delivery devices on SM-induced skin and ocular lesions.

## 1. Introduction

The need for efficient dressings to improve the healing of injuries induced by sulfur mustard (commonly known as mustard gas; 2,2′-dichlorodiethyl sulphide; SM) is still relevant. SM was first used on 12 July 1917 as a chemical warfare agent during World War I [1]. Despite the Chemical Weapons Convention (CWC) ratified by 193 countries [2], there has been evidence or allegation of its use in several recent conflicts, mainly in the Middle East in 2015 [3,4,5]. In addition to the threat of terrorist attacks, SM is a recurrent source of accidental exposure due to unreferenced stocks of artillery shells or buried unexploded shells from World War I [5,6]. Moreover, depending on the climatic conditions, SM can persist in the ground or porous materials for 24 to 48 h in a mild climate or even several months or years in cold and damp conditions [7,8].

SM is a strong vesicant and a bifunctional alkylating agent that damages cells at the molecular level, leading to multiple tissue and cellular targets. Moreover, SM causes chemical burns, and the progression and degree of SM-induced injuries are dependent on the dose, temperature, moisture level, and anatomical site of exposure (skin, eye …) [9,10]. It spreads throughout the body in only a few minutes [11]. In the most severe cases of respiratory failure or bone marrow alteration, death occurs in 1 to 5% of victims after a delay of several days or weeks [3,12]. Very high doses of SM (cutaneous: 64 mg/kg; respiratory: 1500 mg/min/m^3^) can be lethal in around 1 h [3,7]. Until now, there has been no antidote, and medical strategies are still limited to symptomatic and supportive care similar to those used for thermal burns.

Numerous wound dressings are available to treat burns, including gauze, transparent films, foam dressings, hydrogels, hydrocolloids, and hydroconductive dressings. Among these, hydrogels show numerous beneficial properties [13]. The present review aims to show how hydrogel dressings may be suitable for improving the wound healing of SM-induced skin and ocular lesions.

## 2. Symptomatology of Sulfur Mustard Exposure

Exposure to liquid or gas SM is insidious because it is painless but leads to severe, acute, and delayed effects on the skin, eyes, and lungs [3,4] with systemic effects and potential carcinogenic and mutagenic consequences [7,14]. The first clinical manifestations are preceded by an asymptomatic latency period of several hours (2–24 h), depending on the dose, duration, and size of the contaminated area. In addition to the skin, eyes, and lungs, internal organs such as the respiratory tract, reproductive system, central nervous system [15], bone marrow, and gastrointestinal tract [6,16] may be injured. These effects can also be associated with immunological imbalances [7].

The medical examination of 34,000 Iranians exposed to SM during the Iran–Iraq war [1980–1988] revealed 13 to 20 years later, the frequency and severity of SM-induced lesions on their lungs (42.5%), eyes (39.3%), and skin (24.5%) [17]. The eyes are the most sensitive organ to SM exposure, even at low doses, with a toxicity threshold of 12 mg.min/m^3^ (compared with 200 mg.min/m^3^ for skin) [8,18].

### 2.1. Sulfur Mustard-Induced Ocular Injury

Ocular complications occur following direct contact with SM in vapor or liquid form. Eyes are especially sensitive to SM because of the aqueous and mucosal nature of the cornea and conjunctiva and their direct contact with the environment [4,7,19]. In particular, the corneal epithelial cells are extremely sensitive to SM due to their high metabolic activity, vascularization, and cellular turnover [3,4]. The first ocular symptoms and signs appear less than 1 h after exposure and include lacrimation, photophobia, limbal abnormalities, a decrease in visual acuity, and progressive ocular surface discomfort (burning, itching, and redness), which gradually progresses to severe ocular pain [4,8]. In the most severe cases, the deep layers of the cornea and limbal vasculature are affected. SM victims show delayed ocular symptoms with corneal lesions (vesication, ulcerations, opacification, dystrophy, thinning), limbal injuries (ischemia and stem cell deficiency), and necrosis areas [4,7]. In addition, subjects can develop myosis, eyelid inflammation, and conjunctivitis, leading to permanent blindness [3]. In the case of mild lesions, symptoms spontaneously improve within 48 h; the corneal epithelium starts to heal within a few days, and full recovery occurs several weeks later [3,4]. For severe lesions, symptoms improve within 3 to 6 weeks [12]. Otherwise, full recovery can involve neovascularization, which can lead to inadequate healing of the cornea, resulting in opacity and vision decrease or loss [4].

The current treatment for acute SM-induced ocular injury mainly depends on clinical observations [20] and includes ocular irrigations, lubrication, topical antibiotics, and corticosteroids [7]. These treatments prevent secondary infections, reduce ocular pain, and control intraocular pressure. However, topical corticosteroids should be used with caution, as they may predispose the cornea to infection by immunosuppression [21]. The most severe ocular symptoms require pharmacological and surgical interventions with ophthalmic drops, artificial tears, anti-inflammatory (anti-IL6, anti-VEGF, etc.) and immunomodulatory drugs, amniotic membrane and endothelial transplantation, and stem cell transplantation [7,22].

### 2.2. Sulfur Mustard-Induced Skin Injury

After SM contact with the skin, due to its strong penetration power and rapid distribution, this lipophilic substance remains on the surface for 2–3 min. Only 20% rapidly crosses the skin barrier within a few minutes (10–15 min) with an absorption rate of 1–4 µg.cm^−2^.min^−1^ and about 50% dermal absorption and 50% systemic circulation. In the skin, SM is mostly distributed into the epidermis (70%), with the remaining part in the basal membrane and dermis [5]. The first cutaneous symptoms appear with a delay of 2–24 h after exposure; erythema and edema are accompanied by severe itching and burning, especially in moist regions such as the axillae, genital–anal region, and base of the neck [3,7]. The affected areas then progress to inflammation and blister formation after 12–48 h. The minimal dose to induce blister formation is 4 µg/cm^2^ with SM vapor compared with 20 µg/cm^2^ with liquid SM [23], although the blistering mechanism has not yet been clearly identified [24]. Blisters can vary in size from small vesicles to larger bullae containing a non-toxic, light yellow serous fluid [25]. After 3 days, ulcers, full-thickness skin loss, necrosis, and eschar appear after the rupture of the largest blisters. Then, edema and erythema may persist and evolve into dry wounds such as hard necrotic lesions [24] within 4–6 days, and a slough begins to separate. Re-epithelialization begins within 3 weeks [4]. These extensive and incapacitating burns may require several months (up to 6 months) of hospital treatment to heal completely [4,12,26]. Skin lesions can also lead to chronic and long-term residual cosmetic and/or functional sequelae in patients, such as atrophy, chronic urticaria, eczema, keloid, late-onset vesication, local hair loss, pruritis, psoriasis, seborrheic dermatitis, telangiectasis, vitiligo, xerosis [4,7,26], hypopigmentation (original lesion), and/or hyperpigmentation (surrounding areas) scarring, and hypersensitivity to mechanical trauma in the healed areas [4,12].

Furthermore, SM-induced skin lesions are chemical burns, clinically similar to first- or second-degree burns but as debilitating and disfiguring as third-degree burns [3,27]. Compared with thermal burns, these necrotic lesions are slower to heal [4] and have similar complications due to the disruption of the normal skin barrier function and immunodeficiency. Thus, both dysfunctions are major causes of septicemia [3,26] and contribute to a higher susceptibility to mortality [26].

The current standard strategy for the management of SM-induced skin injury is similar to thermal burns. It is modulated according to the location, extension, and severity of the lesions [1,24,28] and consists of an early debridement of the wound and its protection using an occlusive dressing [27,29], particularly when treating large blisters. Care is supplemented with the local or systemic application of anti-inflammatory and anti-microbial drugs [30].

## 3. Treatment of Injuries Using Hydrogels

Topical applications have the advantage of directly treating wounds contrary to the systemic administration of drugs into the vascular circulation, which is not specific to the site and not efficient in blood-deficient wounds such as severe burns and chronic wounds [31]. In order to promote wound healing while being site-specific, it is possible to administer drugs topically by loading them into a hydrogel. Moreover, hydrogel dressings are commonly used in a variety of biomedical products, including wound dressings and contact lenses. On account of their specific properties in terms of biocompatibility, tunable porosity, bio-absorbability, reversibility, serialization, and soft rubbery consistency, hydrogel dressings can facilitate and accelerate wound healing [32,33,34,35,36].

### 3.1. Hydrogel Definition

Hydrogels are three-dimensional networks of cross-linked hydrophilic water-swollen polymeric chains. Due to their hydrophilic composition and cross-links between polymer chains, they can reversibly absorb and retain large amounts of water without losing integrity. However, when they are applied as dressings, they are already swollen in water (70–90%) [24], so they can only give out water and cannot absorb much more. They, therefore, can have a low capacity to absorb exudates [24]. The classification of hydrogels is based on several criteria [37,38]. Among these, there is the chain composition (Figure 1). Polymers composing the hydrogels dressings can be naturally derived, such as polysaccharides (alginate, chitosan, hyaluronan, glucan …), glycolipids, proteoglycans, proteins (collagen, gelatin, elastin, fibrin, silk …) and peptides or can be of synthetic origin (polyurethane, poly(N-vinylpyrrolidone (PVP), polyethylene glycol (PEG) …) [39]. The choice of polymer, cross-linking agent, cross-linking agent concentration, and preparation method determine the properties of the hydrogel (e.g., swelling in water, kinetics of biodegradability) and result in a substantial diversity of hydrogels. Indeed, the structure and polymer composition of hydrogels influences the immobilization and release of water-soluble elements such as therapeutic agents, the mixture of nutrients and growth factors, and the migration of cells through the network [33,40]. Furthermore, modulating their physicochemical and mechanical properties allows adjustment of the 3D networks of polymer to mimic best the physical characteristics of the native extracellular matrix (ECM) [41]. Chemically cross-linked hydrogels are more advantageous compared with physically cross-linked hydrogels (gellan gum or alginate) because they offer a better-prolonged drug release and are also more durable than physically cross-linked hydrogels [42].

### 3.2. Healing of Ocular Wounds Using Hydrogels

Due to the confined structure of the eye, the treatment is applied directly in order to counter the limited drug diffusion through the static, dynamic, and metabolic barrier functions associated with systemic circulation [44,45,46]. Moreover, drug efficacy after topical ocular surface application with conventional eye drops is also limited by the high clearance of the tear film (conjunctival blood vessels, lymph vessels, and nasolacrimal drainage) and low drug ocular bioavailability due to the local systemic absorption and blood ocular barriers [44,46,47]. Thus, for ophthalmic wound dressings, hydrogel contact lenses can be used not only for vision correction but also for ocular drug delivery [47]. Indeed, they are ideal drug delivery devices due to their various properties such as innocuity, transparency, anti-microbial barrier, convenient administration, drug release in a controlled and sustained manner at the target site, and shorter time for wound healing [18,47]. As demonstrated in several in vivo studies, hydrogel contact lenses enhance drug-corneal contact time, reduce systemic side effects, require fewer applications, and limit drug loss from the corneal surface control, thereby extending the drug release/action to several days (hydroxyethyl methacrylate (HEMA) hydrogel) or even months (silicone hydrogel) and enhancing their bioavailability to more than 50% [48] compared with 5–10% for conventional eye drops [47]. This effectiveness is particularly beneficial for chronic treatments, as in the case of corneal ulcers or infections [47]. Soaking methods to incorporate the drugs into the hydrogel contact lenses can only incorporate limited quantities of hydrophilic drugs with low molecular weight and can show a “burst” release of therapeutic agents [47].

As previously reported, SM victims suffered from ocular complications in the anterior segment of the eyes, including conjunctivitis and neovascularization. However, hydrogels loaded with epidermal growth factor (EGF) demonstrated that they helped to treat corneal epithelial wounds [42]. Due to its nature as a growth factor, EGF stimulates the proliferation of cells, such as keratinocytes, which help to promote healing. Moreover, nowadays, to treat corneal neovascularization, anti-VEGF (Vascular Endothelial Growth Factor) treatments with intrastromal delivery are required. Indeed, the binding of VEGF to its receptors induces endothelial cell proliferation contributing to neovascularization. Thus, preventing VEGF binding to its receptors suppresses neovascularization. It will be more patient-friendly to use hydrogel contact lenses loaded with anti-VEGF. However, the few pre-clinical trials ongoing are for injectable hydrogel with bevacizumab (an anti-VEGF product used off-label) in combination or not with ranibizumab [49]. Bevacizumab and ranibizumab are very similar substances derived from the same recombinant humanized monoclonal antibody able to bind VEGF. However, due to its design, ranibizumab, compared to bevacizumab, has a smaller size, a higher binding affinity to VEGF, and a lower persistence in the body.

### 3.3. Healing of Skin Wounds Using Hydrogels

For skin wound management, hydrogel dressings also have the property of an “ideal dressing” due to their low cost, biocompatibility, adaptable thickness, repair of the epidermal and dermal composition, and odor control in addition to being easy to apply, remove, prepare, and store [13,32,50]. Hydrogels are mainly indicated for the treatment of deep, dry, and necrotic wounds due to their properties relating to fluid absorption and dead tissue hydration, which facilitate autolytic debridement [13,51] and are thus suitable for the treatment of chronic cutaneous wounds [51]. More precisely, because of their biocompatibility, biodegradability, and similarity to the ECM, naturally-derived polymers appear to be more promising to accelerate the wound-healing process compared with synthetic polymers [52]. However, the polymer composition does not lead to significant differences in the clinical results [32].

Many pre-clinical and clinical studies have successfully demonstrated that hydrogel dressings optimize the speed (50%) [32] and completeness of wound healing by promoting fibroblast proliferation, keratinocyte migration, and epidermal cell replication [53]. Depending on the tissue hydration status, hydrogels can absorb or provide water to the wound environment due to their permeability to gas and water vapor [9,32]. Keeping wounds moist also contributes to providing a mechanical barrier from infections. As non-adherent dressings, they also preserve newly formed tissues, minimize damage extension, reduce pain during dressing changes, and limit the inflammatory response because of the absence of adhesion to the wound, in addition to their analgesic cooling effect [32,50,54]. Hydrogels can cool the skin surface by up to 5 °C for up to 6 h [54]. All these advantages contribute to improving the final appearance of the scar [13,54]. Due to their wide spectrum of medical use, hydrogels are marketed in a large variety of products. However, despite a wide variety of polymers available, current commercial hydrogel dressings are mainly alginate-based (Nu-Gel^®^, Tegagel^®^, Algosteril^®^, Sorbsan^®^, Curasorb^®^) [55]. To cover superficial wounds, it is possible to use hydrogels, not exhaustively, as primary dressings: Nu-Gel^®^, Vigilon^®^, Flexigel^®^, and Aquamatrix^®^. To fill deep wounds, it is possible to use amorphous hydrogels without a fixed shape: Curasol^TM^, Iamin^®^, DuoDerm^®^, Restore^®^ [32], and ActiFormCool^®^ [51].

The direct topical application of hydrogel supplemented with pharmacological agents has been shown to be effective in wound care, both in pre-clinical and clinical settings [31]. For example, in vivo, studies have been performed to accelerate skin wound healing in mice models. Purilon^®^ hydrogel supplemented with platelet-rich plasma leads to significantly faster closure of the wound size from day 3 to day 14. On day 14, there is a significant increase in the average number of blood vessels per area (*p* < 0.001; 6.6 ± 0.2) compared with hydrogel alone (5.6 ± 0.2) and control group (5.2 ± 0.1) but not with platelet-rich plasma alone (6.3 ± 0.2) [56]. Another in vivo study aimed to improve chronic ulcer complications such as diabetic wounds. Polyethylene glycol hydrogel loaded with an angiogenic drug (deferoxamine) was tested on rats with type I diabetes. This dressing was shown to be efficient for repairing diabetic skin wounds in terms of bacterial level and angiogenic activity [57]. From a clinical perspective, treatment using recombinant human platelet-derived growth factor in a topical hydrogel successfully promoted the healing of skin ulcers refractory to treatment for 12 years [31]. Furthermore, to treat chronic skin wounds, dressings can be supplemented with growth factors, anti-microbial agents [58], skin substitutes, or tissue-engineered substitutes (Regranex^®^, Aquacel^®^, Integra^®^, and Graftskin^®^, respectively) [59].

## 4. Management of Sulfur Mustard-Induced Injuries Using Hydrogels

There is still no satisfactory drug to treat SM in a clinical setting. More precisely, due to the absence of an antidote and specific treatment, the main medical strategies are symptomatic and supportive treatments [60]. This situation highlights the importance of further research into the treatment of SM-induced injuries. As reviewed above, hydrogels can be effective during all stages of burn wound treatment [58]. Thus, hydrogel dressings seem to be one of the most relevant dressings to promote wound healing for SM-induced ocular and skin lesions compared with traditional dressings (gauze and compress), adherent dressings, and skin substitutes.

### 4.1. Healing of SM Ocular Wounds Using Hydrogels

To improve SM ocular wound management, one suggested treatment is based on matrix metalloproteinase (MMP) inhibitor because MMP-9 is an enzyme secreted by basal corneal epithelial cells, leading to chronic corneal ulcerations and is upregulated into the cornea after mustard injuries [18,61]. Indeed, doxycycline (DOX) is a systemic MMP inhibitor [62,63]; thus, hydrogel loaded with DOX can be an additional approach. This strategy has been investigated ex vivo on rabbit corneal organ culture models exposed to SM analogs such as half mustard (2-Chloroethyl ethyl sulfide; CEES) and nitrogen mustard (NM) [18]. Anumolu et al. demonstrated using histology that 24 h after exposure, faster edema and neovascularization reduction were observed in vesicant-exposed corneas treated with PEG-based DOX hydrogels compared with a similar dose of DOX delivered in phosphate-buffered saline solution. The authors concluded that DOX in hydrogels accelerates corneal wound healing after vesicant injury due to the prolonged corneal contact time [18]. These observations can be correlated to an in vivo study on a rabbit model. First, Gordon et al. evaluated in vitro four tetracyclines derivatives (DOX, sancycline, 9-t-butyl sancycline, and de-dimethylamino tetracycline) on corneas before applying the most promising candidate (DOX) to the rabbit eye model. The efficacy of DOX on SM-induced ocular injuries was evaluated for dropwise and PEG-based hydrogel delivery by cornea observations and pachymetry (to assess corneal thickness). Indeed, edema can be quantified by an increase in corneal thickness. Treatment with DOX hydrogels had the double advantage of significantly reducing neovascularization (at 28 days; SM + DOX drops: 30%; SM + DOX hydrogel: 48%) and avoiding multiple daily applications to a painful area. However, in exposed animals treated with DOX drops, the edema corrected faster than in those treated with DOX hydrogels (at 28 days; Unexposed: 337 µm; SM: 504 µm; SM + DOX drops: 421 µm; SM + DOX hydrogel: 462 µm) [64]. The authors concluded that this formulation should be further investigated as daily therapy. Improvements to DOX delivery can also be explored. In terms of timing, it was previously shown that the beneficial effect of DOX treatment on SM-induced ocular injuries depends on the injury stage [20,63].

Another potential therapeutic strategy aiming to restore SM-impaired corneal innervation would be to develop hydrogels loaded with anti-VEGF. Anti-VEGF treatment (bevacizumab) delivered by subconjunctival or topical application partially reduced the extent of corneal neovascularization after SM ocular exposure in rabbit models [20].

Among topical treatments for Iranian and Syrian SM victims, dexamethasone and betamethasone have shown beneficial effects in reducing the severity of their ocular injury, such as reducing corneal edema and preventing the development of corneal neovascularization. These steroidal treatments favored the control of acute and chronic inflammation [20]. Iranian SM victims also received topical cyclosporine to improve their dry eye symptoms [20]. However, these active agents have not yet been investigated by being into loaded hydrogels.

Moreover, according to the literature, several biomarkers found in biological fluids (serum, tears …) of SM victims are known to play a role in corneal inflammation, wound healing, and neovascularization. Factors such as tumor necrosis factor α (TNFα), interleukin-1α (IL-1α), interleukin-8 (IL-8), and Fas ligand (FasL) could be considered as potential therapeutic targets. It would also be interesting to evaluate their therapeutic efficacy on SM-induced ocular lesions into loaded hydrogels.

### 4.2. Healing of SM Skin Wounds Using Hydrogels

Due to the above descriptions of the SM-induced injury appearances and the numerous properties of hydrogels, hydrogel dressings themselves appear to be the most adequate to help treat SM-induced skin lesions. Indeed, following wound debridement of necrotic lesions, dressings need to, among others, promote moist wound healing. Furthermore, hydrogels also possess the ability to cool the wound surface, and in vivo studies showed that the cooling of SM-exposed sites by using cool packs might reduce the severity of the resultant lesions [65,66]. These properties, therefore, confirm the idea that hydrogels should be suitable for SM lesions.

Several approaches have been developed to accelerate SM-injury wound healing and, consequently, to reduce the need for long-term care. Among these options, the use of reactive oxygen species scavengers and anti-inflammatory substances has been tested [67]. Currently, hydrogels loaded with DOX have been tested to cure SM-induced skin lesions. In fact, DOX is an antibiotic with pleiotropic therapeutic properties such as reactive oxygen species scavenger and anti-inflammatory and anti-apoptotic agents. It was successfully used in the treatment of inflammatory skin diseases such as rosacea and acne [68]. To improve the wound healing of NM dermal injuries, the strategy of DOX hydrogel has been explored using in vivo studies. The NM-exposed dorsal skin of SKH-1 hairless mice was treated with DOX-loaded (0.25% *w*/*v*) hydrogels 2 h after NM exposure. Skin histology of NM wounds demonstrated the efficacy of DOX compared with untreated or placebo hydrogels. As expected, these results confirmed that DOX hydrogels are a promising approach to relieving mustard-induced skin lesions [9].

As described above, infection is a common complication of SM-induced skin wounds. To prevent undesired microbial contaminations and facilitate the healing of skin lesions, hydrogels with increased antibacterial properties are of great interest. In particular, curcumin is well known for its anti-inflammatory [69], anti-infectious, analgesic, and antioxidant properties. Furthermore, it has been evaluated in a clinical trial conducted on Iranian veterans, and it was found to contribute to improving their quality of life from 35.63 ± 3.85 to 24.78 ± 5.22, *p* < 0.001), their pruritus score (from 40.60 ± 4.43 to 28.23 ± 4.95, *p* < 0.001) and prevent various cancers and malignant lesions [28]. Sandhu et al. selected this molecule and used an in vivo mice model to explore the efficiency of PEG-based curcumin hydrogel in an NM-induced skin burn model. The treatment was applied 3 h after exposure and repeated on days 2 and 3. In this study, the degree of skin inflammation was assessed as a skin burn. The authors demonstrated that curcumin hydrogel significantly accelerated wound closure on a full-thickness excision model. For example, on day 11, they observed a significantly better closure of wounds with the curcumin hydrogel (95.76 ± 7.85%) compared to the free curcumin hydrogel (74.75 ± 3.53%) and positive control (57.85 ± 8.85%). They concluded that the benefits were linked to the down-regulation of the inflammatory response (significant reduction of 184%) and oxidative stress, faster re-epithelialization, and improved granulation tissue formation. Thus, further exploration of the benefits of these curcumin hydrogels should be carried out for SM-induced skin injury.

Further developments for SM-wound skin management could involve hydrogels containing silver nanoparticles [13], antibiotics (vancomycin, tetracycline, ciprofloxacin, and gentamycin) [60], or antioxidant molecules of natural origin such as silibinin [70], which have already shown interesting effects on CEES- and NM-induced acute skin injuries [71,72]. However, these hydrogels have not yet been investigated in the SM context. In the CEES context, a combined formulation containing a solution of antioxidant (acetylcysteine), anti-inflammatory (dexamethasone), PARP inhibitor (nicotinamide nucleotide) agents associated with a loaded hydrogel with pro-cicatrizing (hyaluronic acid, collagen) and antibacterial (doxycycline, silver sulfadiazine) agents was administrated simultaneously and immediately after CEES application and daily for 7 days [73]. Although the conclusion indicates that the hydrogel could be considered effective in the treatment of vesicants-induced skin lesions, additional experiments would be required to support the efficacy of this loaded hydrogel by itself and its efficacy on a wound with a delay applied post-exposure in order to disprove the hypothesis that the results of its combination in the experimental setting, are not induced more by decontaminant than therapeutic effects.

In order to treat skin sequelae as potential dressing, it would also be interesting to evaluate the therapeutic efficacy of hydrogels loaded with Interferon-γ. This cytokine has already shown a beneficial effect on the atopic dermatitis index and quality of life of Iranian veterans treated with subcutaneous injections [28].

## 5. Conclusions

SM is a chemical agent that remains a terrorist threat. Due to the absence of an antidote and specific treatment, the management of SM-induced lesions still represents a challenge, particularly for the skin and eyes. Currently, the treatment of SM-induced injuries is neither standardized nor optimized. Thus, new strategies are needed to accelerate healing and optimize the repair of the function and appearance of damaged tissues.

Numerous studies have demonstrated that due to their structure and properties, hydrogels have many advantages in healing severe burn wounds, thus making them good dressing candidates to help to treat SM-induced skin and ocular lesions. The use of hydrogels has not been fully explored regarding their many uses and the potential molecules that can be tested in this dressing. Thus, much work remains to be conducted. This review clearly shows that hydrogels are excellent excipients that can be supplemented with many active ingredients such as pro-cicatrizing and antibacterial molecules or SM scavengers in order to increase their contact time, distribution, and, thus, bioavailability.

In conclusion, hydrogels seem to be the most appropriate dressings to improve wound healing for SM-induced skin and ocular injuries. The development of innovative solutions using hydrogels is not limited to the treatment of SM- or vesicant-induced lesions but can be more widely applicable to all types of burns.

## Figures and Tables

**Figure 1 biomedicines-11-01626-f001:**
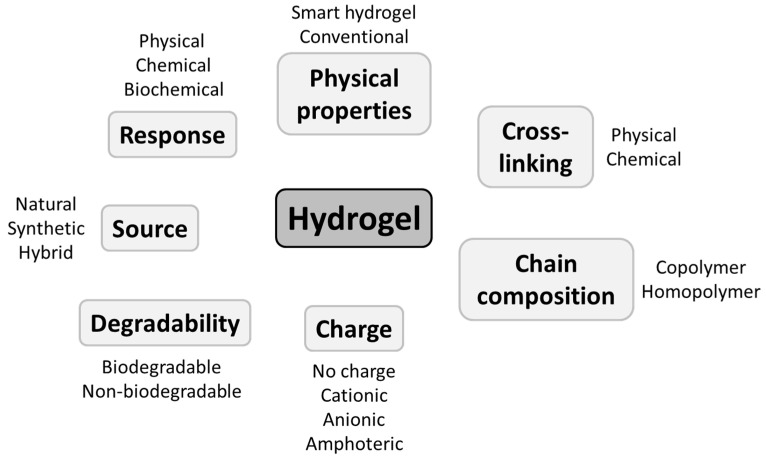
Classification criteria of polymer hydrogels based on [36,43].

## Data Availability

No new data were created or analyzed in this study. Data sharing is not applicable to this article.

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
