# Peer review of "The Use of Hydrogel Dressings in Sulfur Mustard-Induced Skin and Ocular Wound Management"

_biomedicines, 2023, doi:10.3390/biomedicines11061626_

Round 1

Reviewer 1 Report

Caffin and co-workers present a review article concerning the use of hydrogels in trauma due to sulfur mustard exposure. The article is nicely written.

Minor comments:

The authors specifically focus on hydrogels. Maybe they could add a few sentences concerning other biomaterials that might be of potential (e.g. collagen-based materials, alginates etc.).

all in all fine.

Author Response

We would like to thanks you for your encouraging words.

Reviewer 2 Report

This manuscript is a very short review on the use of hydrogels in the treatment of mustard gas-induced skin and ocular lesions. This is a timely and important subject issue since, as mentioned by the authors, there is no antidote for SM. Although there is not much published work in this area, this review does not provide a complete view of the area: (i) it missed some published work, (ii) it did not cover the patent literature (where a few proposals can be found), and (iii) it did not cover the use of hydrogels on their own, being focused on hydrogels containing active substances. As such, it provides an incomplete view of the area and has to be extended. Although well-organized and well-written, in a some cases, there is an excessive use of references and, on occasion, the selected references did not contain what they were supposed to contain, based on the citation. These and other aspects that need revision are detailed in the comments below.

COMMENTS

1. Title: “Interest of hydrogel dressings in sulfur mustard-induced skin and ocular wound management”. Please check if you agree in changing the title to “The use of hydrogels in sulfur mustard-induced skin and ocular wound management”.

2. Abstract: The abstract fails to mention what is reviewed and which was the aim of the review.

3. Section 1, lines 31-39: In this paragraph, the authors mention the effects of sulfur mustard (SM). Then, in the following paragraph, they talk about the use of hydrogels in burns treatment, without clearly establishing a similarity between burns and SM-induced lesions or mentioning that SM causes chemical burns; the only mention is that SM-induced lesions are treated similarly to thermal burns. Although the authors mention this similarity further down in the Introduction, they should also mention it here.

4. Section 3, lines 122-128: In the first sentence, the authors mention topical application of drugs in severe burns. In the following sentence, they seem to continue this subject but, instead, they cover the use of hydrogels in wound healing. As simple hydrogels employed in wound healing cannot be classified as drugs, the authors have to clearly distinguish these two different approaches. Another important approach that is missing and that should be mentioned in this introductory part of section 3, is the combination of both, i.e., hydrogels that release drugs and other active agents to improve wound healing (that are mentioned later in a subsection).

5. Section 3.1, lines 131-133: “Hydrogels are three-dimensional networks of cross-linked hydrophilic water-swollen polymeric chains. Due to their hydrophilic composition, they can reversibly retain large amounts of water (70-96%) [9, 25, 38-40], although they have a low absorbance capacity for exudates [25, 38].”.
a) In the first two sentences above, the authors present a definition of hydrogel. This definition is supported by five references. This is an excessive number of references. Additionally, not all of these references contain this definition of hydrogel.
b) The percentage of retained water that allows a polymer to be classified as a hydrogel varies among authors but it starts below 70% and ends well above 96%. Please review the literature on the definition of hydrogel.
c) The statement that hydrogels “(…) can reversibly retain large amounts of water (…)” should be altered to “can reversibly absorb and retain large amounts of water without losing integrity”, for the sake of completeness.
d) In these sentences, it is stated that hydrogels can retain large amounts of water and that they have a low absorbance capacity for exudate. As exudate is an aqueous solution, this needs clarification. In fact, hydrogels can absorb exudate when they are dry but, when swollen, they either have a more limited capacity to absorb exudate or may donate water. The authors should rewrite this part to clarify this issue.
e) The authors should add a sentence mentioning why hydrogels can absorb and retain large amounts of water.

6. Section 3.1, lines 134-136: “The choice of cross-linking agents and preparation method determine the properties of the hydrogel (e.g., swelling in water, kinetics of biodegradability) and result in a substantial diversity of hydrogels.” There are more factors that determine the hydrogels’ properties but, an important one that must be added, is composition.

7. Section 3.1, lines: 136-140. The authors start this sentence by saying that hydrogels can mimic the extracellular matrix (ECM). However, at the start of the following sentence, they change the subject, mentioning the use of hydrogels in controlled release, to return to the first subject at the end of the sentence. This part must be rewritten to mention how hydrogels mimic the ECM and how they can be used in controlled release of drugs.

8. Section 3.2, line 156. “(…) hydrogel contact lenses with their higher viscosity enhance drug-corneal contact time, (…)”.  One cannot talk about viscosity in the case of a soft contact lens, since viscosity is a property of fluids.

9. Section 3.2, line 160.
a) Ref. 46 should be removed, since it does not mention soft contact lenses (SCLs) in ocular drug delivery.
b) The name “silicon hydrogel” is incorrect. “Silicon” is the name of the chemical element Si. The common name of PDMS (polydimethylsiloxane) is “silicone”.

10. Section 3.2, line 166. Ref. 33 is not about drug-loaded contact SCLs and it should be removed.

11. Section 3.2, line 168. This line contains the first use of the VEGF abbreviation. It must also appear in full.

12. Section 3.1, lines 163-171. The authors here discuss the use of SCLs in ocular drug delivery as it this is the first time that they mention this subject. However, they have already mentioned it in the previous paragraph. Please rewrite this part on the use of SCLs in ocular drug delivery.

13. Section 3.1, lines 166-171. In this sentence, the authors mention the use of EGF-loaded hydrogels to treat corneal epithelial wounds and the use of anti-VEGF drugs to treat macular degeneration. The reader may want to know the relevance of macular degeneration to ocular lesions caused by SM. Additionally, as the acronyms EGF and VEGF are similar, this may cause confusion since the first (EGF) is administered while, the second (VEGF) is inhibited with administered drugs. This part must be rewritten to clarify the role of these two growth factors  and this paragraph and the next should be combined.

14. Section 3.3, line 183: Ref. 25 should be removed, since it is not a primary reference about the mentioned subject.

15. Section 3.3, line 186: There are many more natural polymers employed in wound dressings. The authors should mention those that appear in the cited reference, including proteins.

16. Introduction: As this review covers approaches combining drugs with hydrogels to treat SM-induced lesions, a presentation of the drugs employed in treating SM-induced lesions and their mechanisms of action, in particular, those mentioned in this review, would be helpful.
Section 4.1: In this section, the authors have covered the combination of hydrogels and active substances to treat SM-induced lesions. However, they missed some published work. In fact, a quick bibliographical search retrieved two papers not covered in this review: work by Secara et al. and a work by Anumolu et al. (2011; although this reference appears in the references list, this work does was not mentioned in section 4). In addition, a patent literature search provides a few additional proposals. Finally, given that the title of this review is “Interest of hydrogel dressings in sulfur mustard-induced skin and ocular wound management”, the use of hydrogels on their own must also be covered and was not covered. Actually, there are a few papers on the use of hydrogels in surface decontamination via removal or neutralization under mild conditions. The authors should assess if these approaches could be employed on the skin or on the eye.

17. Section 4.1, line 232: The authors refer to DOX as an “anti-matrix metalloproteinase”. The prefix “anti” is more commonly employed when the active substance is an antibody. As such, DOX should be referred to as a matrix metalloproteinase inhibitor, as the authors do in the following sentence.

18. Section 4.1, line 237: First use of the CEES abbreviation. It must also appear in full.

19. Section 4.1, lines 255-258: This paragraph covers a suggestion of an active substance that could be added to hydrogels to treat SM-induced ocular wounds. However, many others can be proposed. As such, this part is incomplete.

Manuscript is well-written. Minor editing required.

Author Response

We would like to thank you for your detailed and constructive comments.

Reviewer 3 Report

The authors review the hydrogel dressings in sulfur mustard-induced skin  and ocular wound. The manuscript can be published in Biomedicines after minor revision.

The authors give detail information about the symptoms , the different kind of injuries (ocular and skin) and only after 5 pages give details about the hydrogels, and how they can prevent the symptoms after exhibition to SM. The authors are focused mainly to the medicine and how the injuries can be cured. The authors should give more details about the chemical components of possible hydrogels against SM, and they should quote possible biological results from in vitro or in vivo models according to the references.

Author Response

We would like to thanks you for your helpful comments.

Round 2

Reviewer 2 Report

REVIEW # 2
Following the review of this manuscript, many issues raised by this reviewer were corrected, but a few fundamental issues were not addressed or were not satisfactorily addressed. Among these are the lack of information concerning the role of several therapeutic agents mentioned in the review, the swelling range of hydrogels, and a clarification of the title and objective of this review. Additionally, as a result of some of the authors replies, the title of this review may or may not have to be altered.
The new comments are presented below, maintaining the original numbering of the comments, as well as the first comments and replies. All comments that were satisfactorily addressed were omitted; the comments on the authors’ replies were added after the authors’ replies, were labelled “Reviewer’s 2nd comment” and were written in red. Two new comments were added at the end.

 COMMENTS

4. Section 3, lines 122-128: In the first sentence, the authors mention topical application of drugs in severe burns. In the following sentence, they seem to continue this subject but, instead, they cover the use of hydrogels in wound healing. As simple hydrogels employed in wound healing cannot be classified as drugs, the authors have to clearly distinguish these two different approaches. Another important approach that is missing and that should be mentioned in this introductory part of section 3, is the combination of both, i.e., hydrogels that release drugs and other active agents to improve wound healing (that are mentioned later in a subsection).

Author’s response: To clarify the notions addressed in this paragraph (wound healing or severe burns) and to add the notion that hydrogels allow the release of drugs, the section has been completely rewritten: “Topical applications have the advantage of directly treating wound contrary to the systemic administration of drugs into the vascular circulation, which is not specific to the site and not efficient in blood deficient wound as severe burns and chronic wounds [32]. A way of topical administration of these drugs or active agents to improve wound healing is their load into a hydrogel. Therefore, hydrogel dressings are commonly used in a variety of biomedical products, including wound dressings and contact lenses. On account of their specific properties in terms of biocompatibility, tunable porosity, bio-absorbability, reversibility, serialization, and soft rubbery consistence, hydrogel dressings can facilitate and accelerate wound healing [33-37].

Reviewer’s 2nd comment:
a) The word “wound” in line 129 should be replaced by “the wound” or “wounds” and, in line 130, by “wounds”.
b) Lines 131–132: Please rewrite this sentence correctly.

5. Section 3.1, lines 131-133: “Hydrogels are three-dimensional networks of cross-linked hydrophilic water-swollen polymeric chains. Due to their hydrophilic composition, they can reversibly retain large amounts of water (70-96%) [9, 25, 38-40], although they have a low absorbance capacity for exudates [25, 38].”.

b) The percentage of retained water that allows a polymer to be classified as a hydrogel varies among authors but it starts below 70% and ends well above 96%. Please review the literature on the definition of hydrogel.

Author’s response: b) The percentage of retained water kept is [70-90%] from the author #25.

Reviewer’s 2nd comment: Ref. 25 is not the primary reference for the quoted range, since it quotes another reference. In that other reference, the 70–90% range is mentioned for hydrogel dressings, not for hydrogels. As in the first paragraph of this section (section 3.1) the authors are introducing hydrogels – not hydrogel dressings –, this range is incorrect. In fact, polymers that swell less than 70% and more than 90% can still be classified as hydrogels. As requested, please review the literature on this subject and correct this range. Alternatively, as hydrogels are frequently defined without mentioning this range, it can be omitted.

6. Section 3.1, lines 134-136: “The choice of cross-linking agents and preparation method determine the properties of the hydrogel (e.g., swelling in water, kinetics of biodegradability) and result in a substantial diversity of hydrogels.” There are more factors that determine the hydrogels’ properties but, an important one that must be added, is composition.

Author’s response: As suggested, the following information has been added: “… The choice of cross-linking agents, polymer chains composition and preparation method determine the properties of the hydrogel.”

Reviewer’s 2nd comment: Please check if you agree with the following alterations to the added sentence: “The choice of polymer, cross-linking agent, crosslinking agent concentration, and preparation method determine the properties of the hydrogel (e.g., swelling degree in water or biodegradability kinetics) and result in a substantial diversity of hydrogels.”

Point 12: Section 3.1, lines 163-171. The authors here discuss the use of SCLs in ocular drug delivery as it this is the first time that they mention this subject. However, they have already mentioned it in the previous paragraph. Please rewrite this part on the use of SCLs in ocular drug delivery.

Author’s response: To clarify the notions addressed in this paragraph (use of SCL), the section has been  rewritten: “... Thus, for ophthalmic wound dressings, hydrogel contact lenses can be used not only for vision correction but also for ocular drug delivery [49]. Indeed, they are ideal drug delivery devices due to their various properties such as innocuity, transparency, anti-microbial barrier, convenient administration, drug release in a controlled and sustained manner at the target site, and shorter time for wound healing [18, 49]. “

Reviewer’s 2nd comment: Lines 165–167 – It should be added that chemically crosslinked hydrogels are also more durable than physically crosslinked hydrogels (also mentioned in the quoted reference).

13. Section 3.1, lines 166-171. In this sentence, the authors mention the use of EGF-loaded hydrogels to treat corneal epithelial wounds and the use of anti-VEGF drugs to treat macular degeneration. The reader may want to know the relevance of macular degeneration to ocular lesions caused by SM. Additionally, as the acronyms EGF and VEGF are similar, this may cause confusion since the first (EGF) is administered while, the second (VEGF) is inhibited with administered drugs. This part must be rewritten to clarify the role of these two growth factors  and this paragraph and the next should be combined.

Author’s response: to avoid the confusion beetween the use of EGF and VEGF, the section has been  rewritten: “... As previously reported, SM victims suffered of ocular complications in anterior segment of eyes, including conjunctivitis and neovascularization. However, hydrogels loaded with epidermal growth factor (EGF) demonstrated that they helped to treat corneal epithelial wounds [44]. Moreover, nowadays to treat corneal neovascularization anti-VEGF (Vascular Endothelial Growth Factor) treatments with intrastromal delivery are required. It will be more patient-friendly, to use hydrogels contact lenses loaded with anti-VEGF. However, the few preclinical trials ongoing are for injectable hydrogel with bevacizumab (an anti-VEGF product used off-label) in combination or not with ranibizumab [51].

Reviewer’s 2nd comment: This issue was not addressed satisfactorily; as requested, the authors must briefly mention the role of these two growth factors. Additionally, the role of ranibizumab and its relation to bevacizumab has to be mentioned.

15. Section 3.3, line 186: There are many more natural polymers employed in wound dressings. The authors should mention those that appear in the cited reference, including proteins.

Author’s response: As suggested, other potential biomaterials have been developped but in section 3.1, the following information has been added: “…Polymers composing the hydrogels can be naturally-derived such as polysaccharides (alginate, chitosan, hyaluronan, glucan …), glycolipids, proteoglycans, proteins (collagen, gelatin, elastin, fibrin …) and peptides or can be of synthetic origin origin (polyurethane, poly(N-vinylpyrrolidone (PVP), polyethylene glycol (PEG), nanopar-ticles composite-polylmers …)..”

Reviewer’s 2nd comment: It is not clear what the authors meant by “nanoparticles composite-polylmers”.

Point 16: Introduction: As this review covers approaches combining drugs with hydrogels to treat SM-induced lesions, a presentation of the drugs employed in treating SM-induced lesions and their mechanisms of action, in particular, those mentioned in this review, would be helpful.

Section 4.1: In this section, the authors have covered the combination of hydrogels and active substances to treat SM-induced lesions. However, they missed some published work. In fact, a quick bibliographical search retrieved two papers not covered in this review: work by Secara et al. and a work by Anumolu et al. (2011; although this reference appears in the references list, this work does was not mentioned in section 4). In addition, a patent literature search provides a few additional proposals. Finally, given that the title of this review is “Interest of hydrogel dressings in sulfur mustard-induced skin and ocular wound management”, the use of hydrogels on their own must also be covered and was not covered. Actually, there are a few papers on the use of hydrogels in surface decontamination via removal or neutralization under mild conditions. The authors should assess if these approaches could be employed on the skin or on the eye.

 Author’s response:

- Decontamination and treatment are two different strategies. Decontamination prevents vesicants penetration into the body, while the treatments improve / accelerate the wound healing after vesicants penetration. This review is focused on the use of hydrogels for treatment of SM-induced injuries. This is this reason why that references mentioning the use of hydrogels in decontamination processes have not been cited, to avoid confusion.

- Moreover, the work by Secara et al, looks to be more a prophylactic treatment than a curative one. Indeed, it is deposited simultaneously with the vesicant and in parallel with another liquid treatment. It is therefore difficult to determine the effect of the hydrogel loaded with multiple agents on skin lesions and whether it is not linked to a scavenger/decontamination effect as compared to treatment. All the other references in this review related to active hydrogels are added +/- 2h after vesicant exposure, thus allowing time for its penetration.

- As noticed, the reference “18” has been corrected into reference “9  .

Reviewer’s 2nd comment:

a) The authors have not addressed satisfactorily the following raised issue: “As this review covers approaches combining drugs with hydrogels to treat SM-induced lesions, a presentation of the drugs employed in treating SM-induced lesions and of their mechanisms of action, in particular, those mentioned in this review, would be helpful.” This should be done for the mentioned therapeutic agents, but it should be done in a very concise mode; in some cases, mention of the type of drug is sufficient.

b) The authors have not addressed satisfactorily the following raised issue: inclusion in this review of approaches that were published as patents. Despite that patents are not often mentioned in reviews, in many cases due to the fact that they are numerous, in this case, since they are very few, they could be mentioned easily and even employed to highlight the fact that more work on the treatment of SM-induced lesions is needed.

c) Work by Secara et al.: although, as the authors have mentioned in their reply, this work was not designed as a therapeutic study, it could be used to develop a therapeutic approach. In fact, Secara et al. state that their study has therapeutic value and can guide therapy, and there seems to be no reason not to believe this. As such, this work fits in the subject of this review. It should be added to the section where the authors present potential therapeutic strategies (lines 294–297), accompanied by the comments that the authors find relevant.

d) The title of this work is “The use of hydrogel dressings in sulfur mustard-induced skin and ocular wound management”. As such, it would be expected to cover the use of hydrogels as wound dressings on the “management” of SM-induced injuries, both on their own and in combination with other molecules, such as therapeutic agents. Considering this and the authors’ replies and comments, the following comments have arisen:

i) The first approach (use of hydrogels on their own) is not properly covered in this review. Although it is mentioned a few times, mention of studies in which hydrogels on their own were employed amounts to two studies (first paragraph of section 4.2, refs. 64 and 65). However, on close inspection, the two mentioned references do not pertain to studies that employed hydrogels; they are about the effect of low temperature on the exposure sites, in which the low temperature effect was achieved through the use of coolpacks. The authors have then added that hydrogels have a cooling effect and, as such, they could be employed to reduce the wound temperature, but a reference supporting this statement is missing. Thus, these two studies are not examples of the use of hydrogels on their own in the treatment of SM-induced wounds but rather of a potential use. Thus, either the authors add studies in which hydrogels on their own were employed in the treatment of SM-induced injuries (or, in a separate section, elaborate on the potential use of hydrogels on their own in the treatment of SM-induced injuries), or the first part of the title of this review (“The use of hydrogel dressings (…)) would have to be changed to “The use of drug-loaded hydrogel dressings (…)”. Please do not forget that, in case of changes to the title, the whole manuscript would have to be revised in order to adapt it to the new title.

ii) The authors state that they do not intend to cover decontamination strategies based on hydrogels because their review is focused on the use of hydrogels for treatment of SM-induced injuries. Being so, the title has to be changed, since it mentions ”(…) sulfur mustard-induced skin and ocular wound management”, when it would have to mention “(…) sulfur mustard-induced skin and ocular wound treatment”. Please do not forget that, in case of changes to the title, the whole manuscript would have to be revised in order to adapt it to the new title.

19. Section 4.1, lines 255-258: This paragraph covers a suggestion of an active substance that could be added to hydrogels to treat SM-induced ocular wounds. However, many others can be proposed. As such, this part is incomplete.

Author’s response: To complete this section, 2 paragraphs have been added

“Among topical treatments for Iranian and Syrian SM victims, dexamethasone and betamethasone have shown beneficial effects in reducing the severity of their ocular injury, as reducing corneal edema and preventing development of corneal neovascularization. These steroidal treatments favored the control of acute and chronic inflammation [20]. Iranian SM victims also received topical cyclosporine, to improve their dry eye symptoms [20]. However, these active agents have not yet been investigated by being into loaded hydrogels.

Moreover, according to the literature, several biomarkers found in biological fluids (serum, tears …) of SM victims are known to play a role in corneal inflammation, wound healing and neovascularization. Factors as TNFα, IL-1α, IL-8 and FasL could be considered as potential therapeutic targets. It would be also interesting to evaluate their therapeutic efficacy on SM- induced ocular lesions into loaded hydrogels.”

Reviewer’s 2nd comment: The full name of the new abbreviations employed in these sentences is missing.

NEW COMMENTS

20. The single sentence in lines 191–193 should not constitute a paragraph by itself and should be incorporated in the preceding paragraph.

21. Sentence added in lines 234–235: “Despite this availability of materials, commercially hydrogel are mainly based on alginate.”. This statement must be correctly written and must be supported by a reference.

Some of the newly added sentences need to be revised.

Author Response

First of all, we would like to thank you for the thorough examination of our corrected manuscript and your further constructive remarks.

Round 3

Reviewer 2 Report

The last remaining issues have been satisfactorily addressed, with the exception of the addition of the work by Secara et al., in which the reviewer and the authors have different opinions. There is also a comment on the issue concerning the inclusion of patents.
All comments and parts of comments that were satisfactorily addressed were omitted below. The comments on the authors’ replies were added after the authors’ replies, were labelled “Reviewer’s 3rd comment” and were written in red.

COMMENTS

Point 16

Section 4.1: In this section, the authors have covered the combination of hydrogels and active substances to treat SM-induced lesions. However, they missed some published work. In fact, a quick bibliographical search retrieved two papers not covered in this review: work by Secara et al. and a work by Anumolu et al. (2011; although this reference appears in the references list, this work does was not mentioned in section 4). In addition, a patent literature search provides a few additional proposals.

Author’s response:

- Moreover, the work by Secara et al, looks to be more a prophylactic treatment than a curative one. Indeed, it is deposited simultaneously with the vesicant and in parallel with another liquid treatment. It is therefore difficult to determine the effect of the hydrogel loaded with multiple agents on skin lesions and whether it is not linked to a scavenger/decontamination effect as compared to treatment. All the other references in this review related to active hydrogels are added +/- 2h after vesicant exposure, thus allowing time for its penetration.

Reviewer’s 2nd comment:

b) The authors have not addressed satisfactorily the following raised issue: inclusion in this review of approaches that were published as patents. Despite that patents are not often mentioned in reviews, in many cases due to the fact that they are numerous, in this case, since they are very few, they could be mentioned easily and even employed to highlight the fact that more work on the treatment of SM-induced lesions is needed.

c) Work by Secara et al.: although, as the authors have mentioned in their reply, this work was not designed as a therapeutic study, it could be used to develop a therapeutic approach. In fact, Secara et al. state that their study has therapeutic value and can guide therapy, and there seems to be no reason not to believe this. As such, this work fits in the subject of this review. It should be added to the section where the authors present potential therapeutic strategies (lines 294–297), accompanied by the comments that the authors find relevant.

Authors’ 2nd response:

a & b) - In order not to make the introduction too long and disadvantageous the focus on the use of hydrogel dressings for SM injuries, we declined the proposal to go into more detail on the patents and treatments and their mechanisms currently required for the symptomatic treatment of SM injuries. Especially since another reviewer noticed that the prequel to the use of hydrogels for SM lesions is a little too long.

However, to emphasise the importance of further research into the treatment of mustard-induced injuries, we have added the following sentence “This situation highlights the importance of further research into the treatment of SM-induced injuries.”in section 4 – lines 270-271.

Reviewer’s 3rd comment:

a & b) The authors argue that “In order not to make the introduction too long and disadvantageous the focus on the use of hydrogel dressings for SM injuries, we declined the proposal to go into more detail on the patents and their mechanisms (…)”. However, work on treatment approaches to SM-induced injuries does not belong to the Introduction of this review; it is the core of this review (even when the role of the employed therapeutic agents is not mentioned). A patent search with the words” mustard gas” or “sulfur mustard” and “hydrogel*” retrieved only 2 relevant patents on dermal and ocular treatments. As mentioned before, they could easily be added. However, as also mentioned before, it is not obligatory to mention studies published as patents.

Authors’ 2nd response:

c) with all our respect, we prefer to decline the proposal to add the Work by Secara et al in potential therapeutic strategies, for the reason that the protocol described therein does not allow the application of the molecules on a wound.

Reviewer’s 3rd comment:

c) As mentioned in the previous comment about this issue, that work was not designed as a therapeutic study (as such, the described protocols cannot be applied in therapy) but it could be employed to develop a therapeutic approach, as was stated by the authors of that work. If, in the manuscript under review, the authors had presented therapeutic approaches only, their position would be justified. However, the authors have also included potential therapeutical approaches (e.g., lines 286–289; 299–304 and 355–358). As such, this work should also be included as a potential therapeutic approach, with all the coments that the authors find relevant.

Minor editing required.

Author Response

Thank for the thorough examination of our corrected manuscript.
